# User-Centred Design and Development of a Smartphone Application (*OverSight*) for Digital Phenotyping in Ophthalmology

**DOI:** 10.3390/healthcare12242550

**Published:** 2024-12-18

**Authors:** Kishan Devraj, Lee Jones, Bethany Higgins, Peter B. M. Thomas, Mariya Moosajee

**Affiliations:** 1Institute of Ophthalmology, University College London, London EC1V 9EL, UK; kishan.devraj.22@ucl.ac.uk (K.D.); lee-jones@ucl.ac.uk (L.J.); bethany.higgins@ucl.ac.uk (B.H.); peterthomas2@nhs.net (P.B.M.T.); 2Moorfields Eye Hospital NHS Foundation Trust, London EC1V 2PD, UK; 3Francis Crick Institute, London NW1 1AT, UK

**Keywords:** digital phenotyping, digital health, ophthalmology, software development, remote sensing

## Abstract

Background: Visual impairment can significantly impact an individual’s daily activities. Patients require regular monitoring, typically occurring within hospital eye services. Capacity constraints have necessitated innovative solutions to improve patient care. Existing digital solutions rely on task-based digital home monitoring such as visual acuity testing. These require active involvement from patients and do not typically offer an indication of quality of life. Digital phenotyping refers to the use of personal digital devices to quantify passive behaviour for detecting clinically significant changes in vision and act as biomarkers for disease. Its uniqueness lies in the ability to detect changes passively. The objective was to co-design an accessible smartphone app (*OverSight*) for the purposes of digital phenotyping in people with sight impairment. Methods: Development of *OverSight* included stakeholder consultations following principles of user-centred design. Apple iOS software frameworks (HealthKit, ResearchKit, and SensorKit) and a SwiftUI developer toolkit were used to enable the collection of active and passive data streams. Accessibility and usability were assessed using the System Usability Scale (SUS) and feedback following a 3-month pilot study. Consultations with patients informed the design of *OverSight*, including preferred survey scheduling and the relevancy of patient support resources. Results: Twenty visually impaired participants (mean age 42 ± 19 years) were recruited to the pilot study. The average score on the SUS was 76.8 (±8.9), indicating good usability. There was a statistically significant moderate negative correlation between SUS scores and visual acuity in both the better (r = −0.494; *p* ≤ 0.001) and worse eye (r = −0.421; *p* ≤ 0.001). Conclusions: *OverSight* offers promising potential for collecting patient-generated health data for the purposes of digital phenotyping in patients with eye disease. Through further testing and validation, this novel approach to patient care may ultimately provide opportunities for remote monitoring in ophthalmology.

## 1. Introduction

Visual impairment represents a significant public health challenge. The number of people living with visual impairment globally is projected to increase to 1.8 billion by the year 2050 [1]. Compared to other chronic health conditions, the negative impact of visual impairment on the patient is extensive [2], affecting a range of aspects of daily living, including mobility, routine activities, employment, social interactions, and mental health [3,4,5,6,7,8,9,10]. These factors combined have a detrimental impact on independence and quality of life [11,12,13].

In the United Kingdom (UK), many common ophthalmic conditions are chronic, meaning they require lifelong clinical monitoring, typically occurring within the hospital eye services. Traditional methods for assessing and monitoring visual impairment primarily rely on direct clinical evaluation using structural and functional assessments. As a result, ophthalmology departments have become the busiest outpatient service in the UK with approximately 8 million appointments annually [14]. The large caseload of patients requiring hospital monitoring causes demand to outstrip capacity. For example, national surveillance studies indicate that patients are suffering preventable harm due to delays in receiving appropriate care [15].

To address these challenges, recent efforts have focused on harnessing digital innovations to revolutionise the assessment and monitoring of visual impairment [16,17,18,19]. For example, home monitoring of visual function using novel devices has been shown to be viable and comparative to clinic-based assessments in patients with eye diseases such as age-related macular degeneration and glaucoma [20,21]. These innovations aim to overcome the limitations of traditional approaches by facilitating remote monitoring, improving the precision of assessments, and enabling timely intervention. Digital phenotyping is emerging as a potential solution to improve patient care. Digital phenotyping describes the process of collecting data from personal digital devices and/or wearables to quantify an individual’s behaviour and lifestyle in situ, relating this information to their health status [22]. The goal of this approach is to leverage technology to capture details that may be clinically useful biomarkers of disease (e.g., step counts) [23,24,25,26]. However, despite these advancements, there is a notable lack of accessible smartphone applications specifically designed for digital phenotyping in individuals with visual impairment. Existing solutions often do not fully address the usability challenges faced by this population, nor do they effectively utilize patient-generated health data (PGHD) for comprehensive clinical monitoring. In this context, we have co-designed and developed the *OverSight* app, an accessible smartphone application tailored to the needs of people with visual impairment. This approach is particularly pertinent in the context of visual impairment, where eye diseases often progress slowly and insidiously, affecting visual function. For instance, as vision worsens, patients may display observable changes in behaviour and lifestyle, such as reduced mobility, particularly in low-lighting conditions [27,28]. By focusing on accessibility and user engagement, our study advances the application of digital phenotyping in ophthalmology, offering a novel tool with the potential to enhance patient monitoring and inform future care strategies. In the broader context of health, digital phenotyping is showing promising results. This concept was initially used in psychiatry; however, it is now extended across several areas of medicine. For example, a search on clinicaltrials.gov reveals a range of health studies that determine health status through digital biomarkers from connected devices and other apps [29]. Outside of psychiatry and chronic diseases, studies also investigate digital biomarkers for infectious diseases. For example, a recent app study investigated whether personal sensor data, such as smartwatch activity and self-reported symptoms, can detect early signs of infection, including COVID-19 [30]. This study reported that physiological data from smartwatches and activity trackers significantly enhance the ability to distinguish COVID-19-positive symptomatic individuals from negative cases beyond symptom analysis alone, highlighting the potential of digital biomarkers in the broader landscape of healthcare.

The ubiquitous adoption of smartphones and integrated sensors enables the tracking of a wide array of health-related data [31]. This approach creates an opportunity to monitor large streams of PGHD, which can be modelled against disease biomarkers. As a result, patterns in lifestyle and behaviour can be tracked and have the potential to serve as useful surrogate measures of visual status and patient well-being. The *OverSight* app was developed for the purposes of digital phenotyping in ophthalmology, which may have future scope for patient monitoring. The app comprises a suite of features that demonstrate health- and vision-related metrics about the user, including graphical visualisations about lifestyle (e.g., daily steps), scheduled surveys, and links to resource websites. The objective of this study was to co-design and develop an accessible smartphone app for the purposes of digital phenotyping in people with visual impairment, where the outcomes can be used to inform the development of apps designed for research or care purposes in ophthalmology.

## 2. Materials and Methods

Development of the *OverSight* app comprised the following stages: 1. Stakeholder consultation, 2. Development of *OverSight*, and 3. Accessibility and usability testing. The research was conducted at University College London and Moorfields Eye Hospital National Health Service (NHS) Foundation Trust in accordance with the Declaration of Helsinki and was reviewed and approved by the Greater Manchester South Research Ethics Committee (reference: 22/NW/0328). 

Recruitment: For the stakeholder consultations, purposive sampling was used where individuals with any progressive sight loss condition were invited to participate in a focus group from patient clinical services at Moorfields Eye Hospital NHS Foundation Trust. For accessibility testing, individuals with very low or no perception of light vision were recruited from Thomas Pocklington Trust, a national charity that supports blind and partially sighted people. For usability testing, working-age adults diagnosed with an inherited retinal disease were recruited from the principal investigators’ outpatient clinic at Moorfields Eye Hospital NHS Foundation Trust and Retina UK. Inclusion criteria were that any participant should be 16 years of age or older and be able to provide informed consent. Participants were approached by the research team and provided with an information sheet. If the prospective participant agreed to participate, then a consent form was presented for completion. All participants were given onboarding instructions via a telephone call, an email, and/or in person.

### 2.1. Stakeholder Consultation

*OverSight* was developed following key principles of user-centred design, whereby end-users were actively involved early in the design process and decisions were based on the preferences and needs of the target population. A series of stakeholder engagement events were held during initial development to canvass opinions from target users about preferred app functionality and to co-design specific features. An online workshop was undertaken to canvass opinions about the concept of *OverSight* and provided case-use vignettes describing the applicability of digital phenotyping in ophthalmic patient management. The workshop was led by professionals external to the research group from the National Institute for Health Research Biomedical Research Facility. 

Further stakeholder consultation was performed using focus groups with adolescents with visual impairment and their families. The objective was to qualitatively explore the acceptability of digital phenotyping. These findings have been reported in detail elsewhere [32]. Potential features were discussed such as personalised notifications or reminders and mental health tracking. Briefly, both adolescents and parents described their interest in using an app for remote monitoring of eye health, provided there were robust safeguarding procedures to ensure data security. Participants recognised the need for new and innovative approaches to managing the large caseload of patients in ophthalmology services.

### 2.2. Development of OverSight

The *OverSight* app provides the fundamental framework for the collection of both passive and active data streams to enable an understanding of individual-level digital phenotypes. The prototype version of the app was launched through TestFlight, an online beta testing service provided by Apple Inc. that allows developers to test their iOS apps with users [33]. *OverSight* has initially been built exclusively for iOS as this allowed for the prototype to be developed at a faster speed with fewer barriers compared to cross-platform hybrid or web-based approaches, primarily due to lower resource consumption (e.g., Central Processing Unit, Graphics Processing Unit, and battery) and developer access to full features of the mobile device [34,35]. The *OverSight* app can therefore integrate with iOS’s latest features and security measures, ensuring a reliable app that fully utilises the sophisticated capabilities of Apple’s ecosystem, such as communication with the Apple Health app. 

### 2.3. Ethical Considerations

Acknowledging the sensitivity of PGHD and participants’ desire for robust safeguarding procedures during the stakeholder consultations, several measures were implemented to ensure security and privacy. All data collected through the *OverSight* app were encrypted both in transit and at rest (converting it into a secure, unreadable format both when it is being sent and when it is stored), using industry-standard encryption protocols. Access to participant data was restricted to authorised research team members only, and no personally identifiable information was stored on the app. The onboarding procedure included a detailed explanation of data collection, usage, and storage, and participants were informed of their right to withdraw from the study at any time without any impact on their clinical care.

### 2.4. Accessibility and Usability Testing

To maximise the usability and inclusivity of *OverSight* for use across a broad range of patient populations, accessibility was assessed by onboarding 3 participants with ultra-low vision or no perception of light vision. The rationale was that if participants from this category could successfully perform the required functions of the app, it would likely be accessible to other cohorts with visual impairment. The participants were observed in their ability to download the app and register an account, review and accept consent approvals, navigate the user dashboard, and locate and complete surveys, all using accessibility features of the device and app. 

Usability was measured in a pilot study wherein patients were recruited to take part in a 3-month pilot study. The purpose was to assess the feasibility of collecting active data and passive data via the app and gain feedback about the general usability. Usability was assessed using the System Usability Scale (SUS), a tool to reliably measure users’ subjective perceptions of the usability of a system, product, or service [36]. The SUS has 10 Likert-scaled items ranging from 1 (strongly agree) to 5 (strongly disagree). Total SUS scores range from 0 to 100, with higher scores indicating better usability. The SUS has been used widely to study software and hardware usability and has recently been evidenced as a suitable method for evaluating the usability of mobile health apps [37]. Correlational analysis was conducted to assess the relationship between SUS scores and visual function. 

## 3. Results

### 3.1. Stakeholder Consultation

A total of eight individuals with visual impairment took part in the development workshop. Following a description of digital phenotyping, a discussion forum was held relating to participants’ general perspectives on this approach, as well as their preferences and priorities regarding app features. Feedback generated in the workshop was collectively used to inform specific aspects of the development and infrastructure of *OverSight*, including preferences for survey scheduling and reminders. Specifically, short surveys that contained only a few questions were preferred to be collected quarterly to six monthly as a maximum, whereas surveys requiring greater time commitments were preferred annually. The suggested maximum duration of time spent to complete surveys was 20 min. Participants preferred the option for SMS text message notifications when a survey was due, and a single reminder message to complete a survey if the target completion date was not met. If a survey response is due, users are prompted to complete the measure upon opening the app.

The relevancy of patient support resources featured on the app was also assessed. Beyond data collection functionalities, *OverSight* serves as a resource tool to direct users to relevant information about their specific eye condition. Participants in the workshop described the need for resources that were tailored to their specific eye condition, which led to the development of condition-specific information in the support section of the app. For instance, individuals with a genetic eye disease can gain access via *OverSight* to Gene Vision (https://gene.vision, accessed on 27 November 2024), a website providing open access to dynamic information on genetic eye diseases [38]. Having access to health- and medical-related information was described as a priority during stakeholder consultation; thus, links to a range of relevant resources are programmed on *OverSight*, including RetinaUK (retinauk.org.uk) and Thomas Pocklington Trust (www.pocklington.org.uk, accessed on 27 November 2024), both of which are recognised for their support and information dissemination for individuals with visual impairment. Additionally, *OverSight* connects users to the NHS website (https://www.nhs.uk/conditions, accessed on 27 November 2024), a reputable source in the UK offering reliable and authoritative information on a wide range of health conditions. 

### 3.2. Development of OverSight

Apple offers iOS software development frameworks to help developers build modern digital health applications. These frameworks are provided through open-source methods where code is made available to directly access features of Apple devices. The software frameworks used for the development of *OverSight* were HealthKit, SensorKit, and ResearchKit [39]. HealthKit is a centralised health and fitness data repository from iPhones and other connected sensors and devices. SensorKit provides access to select raw data or metrics that the system processes from a sensor, such as keyboard metrics. Requests to utilise SensorKit were reviewed and approved by Apple for *OverSight* to integrate with specific elements, which enabled access to data pertaining to lifestyle and behaviours, such as the environment, engagement, and activity of the user. The full list of passive data streams collected via SensorKit is shown in Table 1, as well as the supporting evidence for its collection. ResearchKit was initially used for active data collection, including scheduled surveys and patient-reported outcome measures for the evaluation of self-reported data, such as quality of life. The short surveys embedded in the app were the EQ-5D (a standardised measure of health-related quality of life) and the ONS-4 (a concise measure of personal well-being). Longer surveys requiring relatively more minutes to complete included the Michigan Retinal Degeneration Questionnaire (MRDQ), the National Eye Institute Visual Function Questionnaire (NEI-VFQ), and the Hospital Anxiety and Depression Scale (HADS).

Bringing together these features, the *OverSight* app was built using the Swift programming language within the Xcode Integrated Development Environment. Swift is an efficient and robust general-purpose programming language, which was used due to its versatility and integration with the Apple software ecosystem [48]. The user interface was built on a cloud-based infrastructure due to its scalability, reliable data storage, security measures, and integration of other relevant tools such as analysing software to enhance the application’s purpose (Figure 1). Firebase was integrated directly into the *OverSight* app using the iOS software development kit. Firebase is a comprehensive mobile and web app development platform developed by Google and run on Google’s Cloud Platform. It offers a wide range of tools and services designed to streamline various aspects of app development and provides developers with appropriate tools to build high-quality applications, both for web and mobile, without managing complex infrastructure or backend systems [49], allowing for expedited responses to alterations that are made through feedback, bug fixes, and other iteration requirements. Here, we provide a brief summary of the cloud computing features to form the app infrastructure: Firebase Authentication, used for secure user authentication and completion of consent procedures; Firebase Cloud Firestore, a real-time database providing secure and scalable storage for user data with access controls and authorised permissions; Firebase Security Rules, which enables access control; Firebase Performance Monitoring and Crashlytics, which provides metrics for app performance and stability, allowing for a proactive response to potential security vulnerabilities; Firebase Analytics, used to gather and aggregate usage data; Google BigQuery, a data warehouse to streamline analysis with options for built-in machine learning for all data types; and Twilio, a third-party cloud communications tool that allows participants to receive notifications through SMS text messaging when surveys are due.

A simple user interface was created that embedded the following features to fulfil the purposes of the app:Registration. Upon downloading the app for the first time, users are asked to authorise or decline the app to integrate with health and sensor data sources, a feature enabled within the HealthKit and SensorKit frameworks. At this stage, users are able to review what data will be collected for the purpose of this study. For security purposes, it is not possible for app users to register for an account with *OverSight* until invited by the research team. A unique code is assigned to recruited participants, which is entered by the user during the registration process.Dashboard. The dashboard serves as the landing page following logging into the app. The user interface features the date and any outstanding surveys to be completed. A tab is available where participants can log any notable events or changes in their health in a free text box. This section was included to allow participants the opportunity to provide additional context about their lifestyle and behaviours.Profile. This section gives users access to eye care resources, such as condition-specific charity services and educational webpages. For example, for patients with inherited eye diseases, information is provided about ongoing clinical trials and research, as well as details about eye disease inheritance patterns. Users can access their settings, which include their unique user ID, their registered condition, and the ability to delete or logout from *OverSight*. In addition, contact details for the research team are available.Insights. Users can view their own self-generated data in graphical visualisations. Visualisations present data with a visual component (e.g., using different colours or sizes) and were added as an option to leverage users’ visual skills, while being less reliant on vast amounts of numerical data. These visualisations are available through the app for individuals who are interested in their measurements and can interpret and understand the data that reflect their daily activity and lifestyle patterns.

### 3.3. Accessibility and Usability Testing

Accessibility was assessed with four individuals with no light perception vision (i.e., total blindness) recruited from Thomas Pocklington Trust. Participants downloaded *OverSight*, accessed various features, and responded to example surveys on the app. These participants used the iOS VoiceOver feature to navigate the app. VoiceOver is a built-in screen reader that describes aloud what appears on the device screen, providing comprehensive information ranging from battery level updates, incoming notifications, and app identification [50]. Feedback from the sessions with blind participants included improving the labelling of buttons and controls within the user interface. This was to ensure that VoiceOver could sufficiently identify relevant sections of the app. To enhance the accessibility of *OverSight* and the overall user experience, it was necessary to incorporate additional custom features, particularly by reducing the actions required of the user to complete surveys. Assistive technologies such as screen readers are optimised when specific labels are given to HTML elements that can be announced to the user, such as buttons where the user is required to select a response (e.g., responses on a survey). Due to limited customisation options available in Apple’s ResearchKit framework, the decision was made to instead administer surveys using a bespoke design using the SwiftUI developer toolkit, wherein accessible features could be provided to help screen reader navigation (Figure 2).

TestFlight was used for app beta testing, which requires users to download an additional app from the Apple store, with further testing instructions before the *OverSight* app could be downloaded. This was a protracted process and caused onboarding to become unnecessarily complex, particularly when users were not supervised. To overcome this issue, *OverSight* was submitted to the Apple store for release review and was accepted for public release (Version 1.0 released on 31 August 2023) for assigned participants. As a result, onboarding new participants was a simpler process requiring fewer steps. 

Accessibility and usability were assessed following a 3-month data collection pilot where patients with visual impairment were asked to download the app and provide feedback on their experience. In total, 20 participants (*n* = 12, males) were recruited with an average age of 42 (±19) years. Most participants were diagnosed with retinitis pigmentosa (*n* = 16) or choroideremia (*n* = 3). One participant had low vision attributed to a macular dystrophy. Average best eye visual acuity was 0.7 (±0.9) logMAR. Participant characteristics are reported in Table 2. 

Participants provided feedback after 3 months of using the app. A total of 15 participants (75%) agreed or strongly agreed that the process of downloading and setting up the app was straightforward, with one participant (5%) disagreeing. Thirteen participants (65%) agreed or strongly agreed that the accessibility of the app met their needs, with two participants (10%) disagreeing. It was noted that some of the disagreements occurred due to the initial burden of performing several more steps to download the app through the TestFlight app; however, this was later resolved as described elsewhere. To gain an understanding of usability, users also provided feedback on the SUS. Responses showed that most participants rated *OverSight* positively. Among the 20 users, the average overall SUS score was 76.8 (±8.9). SUS scores are then converted to a percentile rank to show how the app rates relative to others [51]. The normative average score (50th percentile) is 68, meaning a raw SUS above 68 is better than average and below 68 is below average. In total, 15 (75%) of our respondents scored above the normative average of 68. The overall average score of 76.8 falls within the 70–79% percentile on the SUS, equating to ‘good’ usability according to the SUS scoring thresholds [51]. Analysis was conducted to assess the relationship between scores on the SUS and visual acuity (Figure 3). There was a statistically significant moderate negative correlation between SUS scores and visual acuity in both the better (r = −0.494; *p* ≤ 0.001) and worse eye (r = −0.421; *p* ≤ 0.001), suggesting that those with better visual acuity gave higher usability ratings. However, as shown in Figure 3, even participants with significantly reduced vision in their better seeing eye gave fairly high usability scores. For example, the participant with the lowest vision (ID: 13) provided one of the highest SUS scores. This finding indicates that other confounding factors may influence usability perceptions. To explore additional factors, the relationship between age and SUS scores was assessed. A weak positive correlation, which was not statistically significant, was found (r = 0.211; *p* = 0.40), indicating that age may not substantially impact usability ratings. Spearman’s rank correlation was used due to the small sample size and non-normal data distribution. The analysis did not adjust for potential confounders such as type of visual impairment or if users had to download the TestFlight app.

## 4. Discussion

This study describes a co-design approach to the development of *OverSight*, a smartphone app for the purposes of digital phenotyping in people with eye disease. In addition, we report the accessibility and usability of the app following initial testing with target users. *OverSight* was developed following a comprehensive process of stakeholder consultations adhering to key principles of user-centred design, enabling an understanding of the users, their needs, and preferences to maximise the desirability and usability of the app. The SUS scores were within the ‘good’ range, indicating there were no serious issues with usability, but scope for improvements in future iterations of the app. This was demonstrated by the significant negative correlation between BCVA and SUS scores, which underscores the impact of visual impairment on app usability. This suggests that users with more severe vision loss may encounter greater challenges when interacting with digital health applications, although other external factors may have come into play. For example, some users who reported average or low usability were those who had to download the TestFlight app before the app was launched on the app store, thereby requiring further steps to onboard as described previously. This may also account for the observation that several participants with vision falling within the higher range of the group had borderline or below-borderline usability scores, while one participant with the poorest vision achieved one of the highest usability scores. In addition, the limited sample size may have restricted the statistical power. Therefore, further testing could explore a larger sample size to address other confounding variables, outside of age, which had a weak positive correlation, such as the type of eye condition or visual impairment.

The principles of user-centred design recognise the need to actively involve users throughout the development process. In addition, early prototyping and continuous iteration enable an evaluation of design solutions [52]. Our activities included stakeholder consultations and feedback mechanisms, providing a low-cost opportunity to identify target user preferences and a general understanding of the purposes of the app. Participants in the workshop gave suggestions for specific app features, such as survey scheduling, reminders, and tailored patient resources. These recommendations were actioned and, based on responses to the SUS, appeared to be effective in optimising app design in an early stage of development. These results highlight the value of adopting a co-design process during app development but also highlight challenges. For example, while providing tailored patient resources for all users is a key target, patients may have complex or rare eye conditions, some with multiple health and ocular comorbidities. As such, developing a truly personalised app requires consideration of the appropriateness of patient resources and tailoring these as best possible to the specific end-user. Yet, a number of recent developments show promise in improving support and information provision to patients with eye diseases, such as gene.vision [36], a web-based resource for patients with rare genetic eye diseases. Moving forward, it will be essential that *OverSight* remains up-to-date with credible and accessible resources to maintain its usability and desirability among target users. Participants also made clear their expectations regarding data security. This was reinforced during the onboarding process, where an explanation was given on how we satisfied safeguarding procedures, whether it was in-person, over a call, or through participant information sheets.

Digital phenotyping is emerging as an innovative approach to patient monitoring across a range of medical specialities. In psychiatry, the mindLAMP app has successfully been used to predict the risk of relapse in patients with schizophrenia by monitoring anomalies in smartphone passive data streams combined with surveys [53,54]. Similarly, the measurement of mobility and sociability features using the Beiwe app has been used to identify early warning signs of relapse for schizophrenia patients through behavioural anomaly detection [55]. Specifically, the authors found that the rate of behavioural anomalies detected in the 2 weeks prior to relapse was 71% higher than the rate of anomalies during other time periods. Real-time monitoring of digital data streams can serve as powerful biomarkers to aid the detection of relapse risk in patients with serious mental illnesses, creating opportunities for timely intervention and support, with the scope to reduce the cost of care and burden on healthcare services. Beyond psychiatry, digital phenotyping has also been employed in oncology to assess postoperative recovery [56], in endocrinology to monitor glucose response in type 2 diabetes [57], and in orthopaedics to track mobility and pain in patients with spinal disease [58].

The purpose of *OverSight*, and digital phenotyping more broadly, aligns with the growing attention regarding the use and value of PGHD. Digital PGHD can be defined as electronic, health-related data created, directly recorded, or gathered by or from patients outside of a clinical environment, which might include self-reported data on symptoms, activities of daily living, and lifestyle factors, as well as sensor data on physiological or behavioural biometrics [59]. In the UK, the NHS Long-Term Plan is a 10-year strategy describing major practical changes to make better use of data and clinical technology to transform and revolutionise healthcare delivery [60]. Given the ongoing and projected capacity challenges in eye care, the field of ophthalmology represents an ideal candidate for digital innovation. Whilst at present, the adoption of digital phenotyping in eye care remains a considerable way away from widespread adoption; apps such as *OverSight* represent an opportunity to accelerate the active involvement of patients in their care, as well as future scope to routinely monitor chronic diseases in situ, improving the health outcomes of current and future patients.

Other apps have been developed in the field of ophthalmology that have aimed to improve quality and access to care [61,62,63,64,65,66,67,68]. The general focus of research in this area has been directed at creating app-based versions of tests that would usually take place in clinics, such as visual acuity and contrast sensitivity. Evidence suggests that it is feasible to use these apps for home monitoring, with many demonstrating high reproducibility, accuracy, and reliability compared to gold-standard assessments [62,65,66,67,68]. Outside of visual acuity, apps such as Alleye have been shown to be effective in detecting metamorphopsia in patients with AMD [66,67]. However, challenges with adherence are a common issue, highlighting the need for approaches that optimize consistent adoption. Additionally, the responsibility to perform the test accurately is shifted from the provider to the patient. In contrast, *OverSight* uses passive data already collected in situ through the smartphone device, thereby reducing the requirement of the user. Unlike apps that predominantly rely on specific task-based assessments, *OverSight* aims to explore how PGHD can be used as surrogate measures of vision and health status, which can be further explored as potential disease biomarkers. In that respect, *OverSight* offers more than only a single measurement in time and can provide a more holistic approach to understanding the complex dynamics and impact of visual impairment.

Aside from routine patient monitoring, digital phenotyping has promising applicability in other aspects of clinical care and research in ophthalmology. For example, a plethora of phase 3 clinical trials are underway, testing novel therapeutics for a vast number of eye diseases, including inherited retinal diseases. There is ongoing discussion between clinical taskforces and regulatory bodies regarding appropriate outcome measure selection in these trials, as any metric must be functional and meaningful for the patient [69]. Digital phenotyping represents an opportunity to measure the impact of novel therapies by observing real-world changes in behaviour and daily functioning, thus assessing outcomes that are meaningful to the patient themselves. There is scope for app-derived data to be correlated with retinal imaging of pathological changes to the structure and function of the eye over time. This opens the opportunity to utilise large language models (LLMs), a type of artificial intelligence algorithm that uses deep learning techniques and large datasets to understand, summarise, generate, and predict new content. LLMs are becoming an increasingly popular area of interest within ophthalmology, where studies have shown their ability to improve clinical outcomes [70,71,72]. Some of these models have multi-modal capabilities in that they can handle various types of biomedical data, including clinical language and medical imaging [73]. The success of LLMs in other medical fields suggests promising potential for their application in ophthalmological care, particularly in processing the rich data generated through a platform such as Oversight. For example, it could understand large amounts of PGHD from the app to make predictions on what resources, advice, or guidance would be most useful for the patient’s individual circumstances. However, the integration of LLMs into healthcare comes with challenges. Ensuring the accuracy and reliability of AI-generated content is critical, given that errors in medical advice can have serious consequences. There is also the risk of biases in AI models, which can lead to disparities in care [74]. When incorporating digital advancements such as these into *OverSight* and in clinical practice, it is important to also note that varying levels of digital literacy exist, especially amongst the elderly who have the highest risk of eye disease [75,76]. Although usability testing demonstrated that age did not affect the usability of the *OverSight* app, the sample size was small and therefore more work needs to be done to assess its usability amongst the elderly. If not addressed, it can cause a digital divide and therefore increase inequality rather than solving it. Even if considered safe and effective, existing challenges remain to ensure interoperability across the healthcare ecosystem [77]. Therefore, for digital health apps such as *OverSight*, ethical considerations and regulatory approvals are ongoing areas of development, necessitating collaboration between technologists, clinicians, and policymakers to establish appropriate guidelines and standards [78].

One of the main limitations is that the *OverSight* app is currently limited to iPhone users only and it remains in question whether the findings and usability might be applicable or transferrable to other platforms, which restricts the broader applicability of the research. *OverSight* reads health data from Apple’s Health app, and a similar infrastructure on other platforms would require reading data from Google’s Health Connect app connected to its device sensors, which may differ in sensitivity to readings of the Apple Health app. Presently, developers for other platforms, such as Android, do not have the ability to access potentially relevant data streams, such as keyboard metrics, ambient light, or device usage characteristics, thereby excluding certain populations of users within the research or real-world applicability of the app. However, external devices or sensors connected to the *OverSight* app have the potential to mitigate these limitations in the future.

The concept of digital phenotyping has an inherent dependence on the user’s interactions with their smartphone device and associated wearables, resulting in large complex datasets requiring appropriate analytical methods to draw robust interpretations. Interactions can change due to factors unrelated to health status or visual functioning, such as evolving personal habits or external influences. As digital phenotyping continues to mature, there will be a need to understand how to maximise the utility of this approach in a healthcare setting. As described elsewhere, key statistical considerations such as handling missing data and quantifying uncertainty of the estimates will become an area of specific scientific questioning and priority in the near future [24]. 

## 5. Conclusions

A smartphone app, *OverSight*, was co-designed and developed for the purposes of digital phenotyping in patients with eye disease. A user-centred process included active engagement with stakeholders throughout the development journey, followed by pilot testing to determine usability. *OverSight* can be used as a patient resource, providing disease-specific information and access to online patient support. The development of *OverSight* highlights the importance of addressing ethical considerations, including patient privacy and data security. Future research should focus on validating the app with larger and more diverse populations to ensure its usability and effectiveness across different demographics. Additionally, studies assessing the app’s accuracy in detecting clinically significant changes in visual function are essential. Steps toward regulatory approval, including adherence to health data security standards and clinical efficacy trials, will be necessary for integration into routine clinical practice. Incorporating advanced technologies such as LLMs can further personalise patient support and improve the analysis of PGHD. By addressing these aspects and through further testing and validation, studying the relevant data streams collected by *OverSight* will enable the detection of digital biomarkers that are sensitive to changes in the patient’s behaviour and lifestyle, which can be attributable to changes in visual functioning. This insight could inform the foundation of custom sensor data collection, providing the basis for remote monitoring of eye disease beyond the busy ophthalmology outpatient services.

## Figures and Tables

**Figure 1 healthcare-12-02550-f001:**
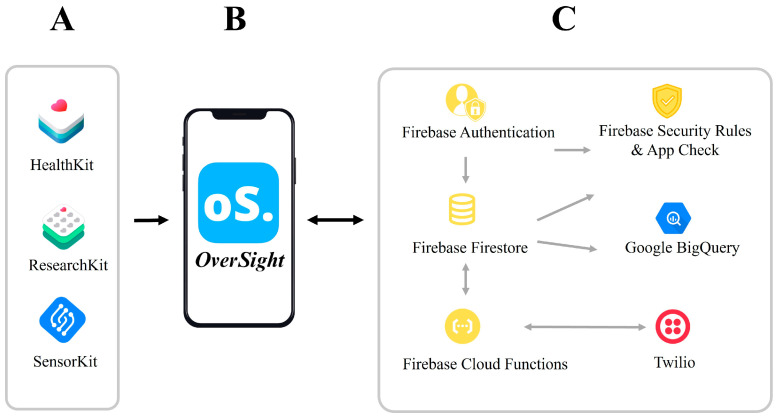
An illustration of the tech stack used to create the *OverSight* app. (**A**) Apple software frameworks. (**B**) *OverSight* app front end. (**C**) Cloud infrastructure.

**Figure 2 healthcare-12-02550-f002:**
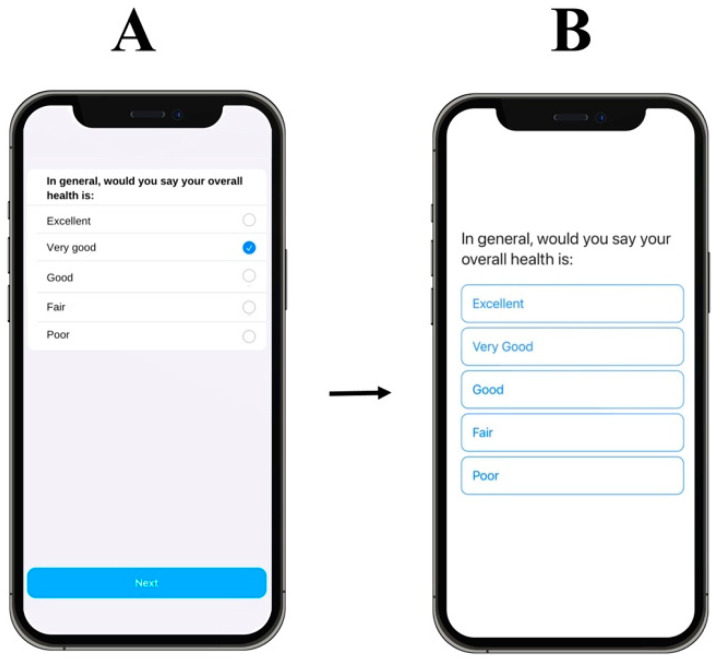
Improved accessibility of surveys. (**A**) Survey developed using the ResearchKit framework, where the user is required to perform multiple actions to submit a response. (**B**) Survey developed using the SwiftUI toolkit, where fewer actions are required by the user and there is enhanced VoiceOver accessibility.

**Figure 3 healthcare-12-02550-f003:**
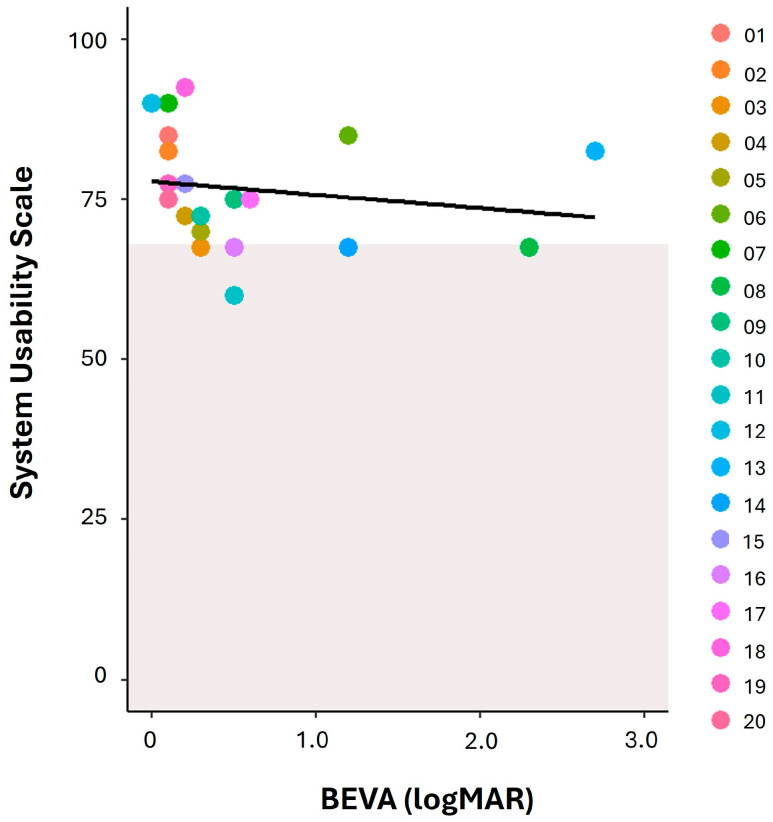
Scatterplot showing the relationship between better eye visual acuity (BEVA) in logMAR and scores on the System Usability Scale (SUS). Each point represents an individual participant. The grey-shaded area represents the normative 50th percentile for the SUS, which is considered the threshold for good usability. The majority of data points (*n* = 15) lie above this area, indicating most participants rated *OverSight* as having better than average usability.

**Table 1 healthcare-12-02550-t001:** Passive data streams collected on *OverSight*.

Apple Framework	Parameter Collected	Description of Parameter	Reasons to Support Parameter Collection
HealthKit	Step count	A quantity sample type that measures the number of steps the user has taken.	Individuals with visual impairment take fewer steps and have reduced mobility than those with normal sight [40,41].
	Walking speed	A quantity sample type that measures the user’s average speed when walking steadily over flat ground.	Visual impairment is associated with slower walking speed compared with age-matched controls [42].
	Distance walking or running	A quantity sample type that measures the distance the user has moved by walking or running.	Reduction in physical activity and walking are associated with greater levels of visual field loss [41].
	Number of times fallen	A quantity sample type that measures the number of times the user fell.	Impaired vision is an independent risk factor for falling and fractures [43].
SensorKit	Ambient light	The amount of ambient light in the user’s environment.	Challenges with daily activities are more pronounced under low-luminance conditions [44].
	Device usage	The frequency and relative duration that the user uses their device, particular Apple apps or websites.	Device usage can be used to measure extent of interactions with smartphones. SensorKit allows for analysis of phone usage and temporal anxiety levels [45], a condition that is common amongst the visually impaired [46].
	Keyboard metrics	The configuration of a device’s keyboard and its usage patterns.	Sentiment analysis through words typed on a keypad can inform about a user’s mental state. A review highlighted the risk of depressive symptoms in individuals who have visual impairments [47].

**Table 2 healthcare-12-02550-t002:** Pilot study participant demographics. For each participant, this includes age bracket, principal diagnosis, for example, Retinitis Pigmentosa (RP), Best Corrected VA Right Eye (BCVA RE), Best Corrected VA Left Eye (BCVA LE), and System Usability Score (SUS).

Participant ID	Age Bracket (Years)	Principal Diagnosis	BCVA RE (logMAR)	BCVA LE (logMAR)	SUS
01	16–30	RP	0.2	0.1	85
02	16–30	Choroideremia	0.1	0.1	82.5
03	31–40	RP	0.3	0.6	67.5
04	16–30	RP	0.2	0.4	72.5
05	51–60	RP	0.3	0.3	70
06	71–80	Macular dystrophy	1.2	1.2	85
07	51–60	RP	0.1	0.2	90
08	41–50	RP	2.3	2.3	67.5
09	41–50	RP	0.5	0.6	75
10	41–50	RP	0.7	0.3	72.5
11	41–50	RP	0.7	0.5	60
12	61–70	RP	0	0.5	90
13	61–70	RP	2.7	2.7	82.5
14	61–70	RP	1.2	1.2	67.5
15	16–30	Choroideremia	0.2	0.5	77.5
16	16–30	Choroideremia	0.5	0.6	67.5
17	51–60	RP	1.7	0.7	75
18	61–70	RP	0.3	0.2	92.5
19	71–80	RP	0.1	0.3	77.5
20	31–40	RP	0.1	0.2	75

## Data Availability

The raw data supporting the conclusions of this article will be made available by the authors on request.

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
