# Peer review of "User-Centred Design and Development of a Smartphone Application (OverSight) for Digital Phenotyping in Ophthalmology"

_healthcare, 2024, doi:10.3390/healthcare12242550_

Round 1

Reviewer 1 Report

Comments and Suggestions for Authors

Dear Authors,

I reviewed your paper with great interest, and found it enjoyable to read.

Major: My main suggestion is to reconsider the reporting of the co-design approach and data collected. I note you say it is reported in detail, elsewhere. In that case (in my view), this paper should solely be about the survey results and explaining the co-design part should go. Also, the survey data is reported descriptively, there are no inferential calculations or triangulations made about the survey items and participant characteristics. I realise this may be due to the small sample size but some explanation for this would be valuable. 

Some minor comments: 

1. It would be good to have a bit more information on recruitment - who did the recruitment, what as the modality of recruitment - e.g. flyer or verbal invitation. What onboarding was provided to participants?

Best wishes, I hope you find these comments helpful.

Author Response

Comments 1. Major: My main suggestion is to reconsider the reporting of the co-design approach and data collected. I note you say it is reported in detail, elsewhere. In that case (in my view), this paper should solely be about the survey results and explaining the co-design part should go.

The reference to reporting details elsewhere was regarding our previous work which assessed the acceptability of digital phenotyping amongst people living with visual impairment. This previous work focused on patient and carer attitudes towards the concept of digital phenotyping, rather than specific design features to include on the app. The present manuscript specifically outlines the co-design and development of the app. For example, the stakeholder consultation assisted in designing features such as personalised notifications/reminders to complete surveys. In addition, a critical aim of the research was to develop an app which is accessible for people with visual impairment. This was done through consultation with patients with ultra low vision who provided important feedback on accessibility functions that could be used to improve usability. For example, the inclusion of features which are compatible with screen reading software. These are the reasons why the co-design and co-development components of this paper are necessary.

Comments 2. Also, the survey data is reported descriptively, there are no inferential calculations or triangulations made about the survey items and participant characteristics. I realise this may be due to the small sample size but some explanation for this would be valuable. 

We had included a statistical analysis assessing the relationship between scores on the system usability scale (SUS) and visual acuity. We agree that our sample is too small to conduct a sufficiently meaningful analysis based on participant demographics, however, we have now included exploratory statistical analysis on the relationship between age and SUS scores indicating whether age has a substantial impact on usability ratings. We have stated the statistical methods used and offered more explanation of these results besides the small sample size:

Line 366: “There was a statistically significant moderate negative correlation between SUS scores and visual acuity in both the better (r = -0.494; p = <0.001) and worse eye (r = -0.421; p = <0.001), suggesting that those with better visual acuity gave higher usability ratings. However, as shown in figure 3, even participants with significantly reduced vision in their better seeing eye gave fairly high usability scores. For example, the participant with the lowest vision (ID: 13) provided  one of the highest SUS scores. This finding indicates that other confounding factors may influence usability perceptions. To explore additional factors, the relationship between age and SUS scores was assessed. A weak positive correlation which was not statistically significant was found (r = 0.211; p = 0.40), indicating that age may not substantially impact on usability ratings. Spearman’s rank correlation was used due to small sample size and non-normal data distribution.”

These findings were elaborated on in the discussion as shown here:

Line 394: “The SUS scores were within the ‘good’ range, indicating there were no serious issues with usability, but scope for improvements in future iterations of the app. For instance, the negative correlation between BCVA and SUS scores suggests that users with more severe vision loss may encounter greater challenges when interacting with digital health applications. Nevertheless, participants with significantly reduced vision still returned notably high scores on the SUS and participants with vision in the top 20% of the group (based on best-eye visual acuity) received borderline or below-borderline usability scores. This demonstrates that other factors outside poor vision may be causing reduced usability, for example potential differences in individual phone accessibility settings. Therefore, further testing could use a larger sample size to address potential confounding variables, such as smartphone accessibility features used and type of eye condition.

Comments 3. Some minor comments: 

It would be good to have a bit more information on recruitment - who did the recruitment, what as the modality of recruitment - e.g. flyer or verbal invitation. What onboarding was provided to participants

Done: We have expanded the recruitment section in the methods explaining how recruitment/onboarding was carried out:

Line 120: “Inclusion criteria was any participant should be 16 years of age or older, and  be able to provide informed consent. Participants were approached by the research team and provided the information sheet. If the prospective participant agreed to participate then a consent form was presented for completion. All participants were given onboarding instructions via telephone call, email and/or in person”.

Reviewer 2 Report

Comments and Suggestions for Authors

Thank you for the opportunity given to me to review this paper. Please find below my review result.

Title and Abstract: The title accurately reflects the content of the paper, making it clear that the focus is on the design and development of a smartphone app for digital phenotyping in ophthalmology. The abstract is concise and provides a good overview of the objectives, methods, results, and conclusion. However, it could benefit from a clearer statement of the novelty and significance of the study, specifically addressing how this app is distinct from previous approaches in digital health. Including specific details on the clinical implications could strengthen the abstract.

Introduction: The introduction effectively sets the stage by providing a background on the importance of visual impairment as a public health issue and the capacity constraints in eye services. The transition to digital health solutions is logical and well-justified. However, the literature review on digital phenotyping could be more comprehensive by incorporating more recent advancements and evidence of its application in healthcare settings beyond psychiatry and other chronic diseases. Expanding the scope slightly would help to contextualize this study within the broader digital health landscape.

Methods: The methods are detailed, but some aspects need more clarity. The stakeholder consultation process is described, but it would benefit from more specifics on how participants were selected and the criteria for inclusion. For instance, were there any exclusion criteria besides visual impairment? The accessibility and usability testing sections are thorough, but the discussion of the ethical considerations (e.g., patient data security) could be expanded, especially considering the increasing concerns regarding patient-generated health data (PGHD).

Furthermore, the authors could provide more information about the statistical methods used to analyze the data, particularly the correlation analysis between System Usability Scale (SUS) scores and visual acuity. Were any adjustments made for potential confounders? Including more details on how the app’s user interface was iteratively improved based on the testing feedback would also be helpful.

Results: The results are presented clearly, particularly in the breakdown of participant demographics and usability scores. However, while the correlation between visual acuity and SUS scores is statistically significant, the clinical relevance of this finding is not thoroughly discussed. It is also worth noting that the sample size of 20 participants is relatively small for drawing strong conclusions about usability and generalizability. More emphasis should be placed on this limitation. Additionally, reporting more detailed findings from the stakeholder consultation could provide greater insight into how specific user needs were addressed in the app's development.

Discussion: The discussion is well-organized and presents a balanced view of the strengths and limitations of the study. The authors successfully argue that the OverSight app could fill a gap in remote ophthalmological monitoring. However, there is room to delve deeper into the potential challenges of integrating such technology into routine clinical practice, such as the digital divide and varying levels of digital literacy among patients.

Moreover, the comparison to other digital phenotyping applications in ophthalmology is useful, but the discussion could benefit from a more detailed comparison with other types of health-related apps to highlight what sets OverSight apart. The mention of future developments using Large Language Models (LLMs) is forward-thinking, but speculative; more discussion on the current state of LLMs in healthcare would ground this section better.

Conclusion: The conclusion summarizes the key findings and future directions well. However, it could be strengthened by including more specific next steps for research and development, such as validation studies with larger, more diverse populations, or steps toward regulatory approval for use in clinical practice.

Overall Recommendations:

  1. Major Revision: Expand the discussion on ethical considerations regarding PGHD and patient data security. Provide additional details on the stakeholder consultation and how user feedback was integrated into the app’s design.
  2. Minor Revision: Improve clarity in the methods section, particularly in describing statistical analyses and selection criteria for participants.
  3. Strengths: The paper’s methodology is well-structured, with a clear focus on user-centered design and thorough testing. The authors present a balanced discussion of the app's potential and limitations.
  4. Weaknesses: The small sample size limits the generalizability of the results. The novelty of the app compared to existing technologies could be more explicitly emphasized.

Author Response

Comments 1. Title and Abstract: The title accurately reflects the content of the paper, making it clear that the focus is on the design and development of a smartphone app for digital phenotyping in ophthalmology. The abstract is concise and provides a good overview of the objectives, methods, results, and conclusion. However, it could benefit from a clearer statement of the novelty and significance of the study, specifically addressing how this app is distinct from previous approaches in digital health. Including specific details on the clinical implications could strengthen the abstract.

Done: See line 14, “Existing digital solutions rely on task-based digital home monitoring such as visual acuity testing. These require active involvement from patients and do not typically offer an indication of quality-of-life. Digital phenotyping refers to the use of personal digital devices to quantify passive behaviour for detecting clinically significant changes in vision and act as biomarker for disease.”

Comments 2. Introduction: The introduction effectively sets the stage by providing a background on the importance of visual impairment as a public health issue and the capacity constraints in eye services. The transition to digital health solutions is logical and well-justified. However, the literature review on digital phenotyping could be more comprehensive by incorporating more recent advancements and evidence of its application in healthcare settings beyond psychiatry and other chronic diseases. Expanding the scope slightly would help to contextualize this study within the broader digital health landscape.

Moreover, the comparison to other digital phenotyping applications in ophthalmology is useful, but the discussion could benefit from a more detailed comparison with other types of health-related apps to highlight what sets OverSight apart.

Done: See line 78: “In the broader context of health, digital phenotyping is showing promising results. This concept was initially used in psychiatry, however it’s now extended across several areas of medicine. For example a search on clinicaltrials.gov reveals a range of studies which determine health status through digital biomarkers from connected devices and other apps. Outside of psychiatry and chronic diseases, studies also investigate digital biomarkers for infectious disease. For example a recent study investigated whether personal sensor data, like smartwatch activity and self-reported symptoms, can detect early signs of infection, including COVID-19. This study reported that physiological data from smartwatches and activity trackers significantly enhance the ability to distinguish COVID-19-positive symptomatic individuals from negative cases beyond symptom analysis alone, highlighting the potential of digital biomarkers in the broader landscape of healthcare.”

Comments 3. Methods: The methods are detailed, but some aspects need more clarity. The stakeholder consultation process is described, but it would benefit from more specifics on how participants were selected and the criteria for inclusion. For instance, were there any exclusion criteria besides visual impairment?

Done: See line 113: Recruitment: For the stakeholder consultations, purposive sampling was used where individuals with a progressive sight loss condition were invited to participate in a focus group from patient clinical services at Moorfields Eye Hospital NHS Foundation Trust. For accessibility testing, individuals with very-low or no perception of light vision were recruited from Thomas Pocklington Trust, a national charity which supports blind and partially sighted people. For usability testing, working-age adults diagnosed with an inherited retinal disease were recruited from the principal investigators’ outpatient clinic at Moorfields Eye Hospital NHS Foundation Trust and Retina UK. Inclusion criteria was any participant should be 16 years of age or older and be able to provide informed consent. Participants were approached by the research team and provided an information sheet. If the prospective participant agreed to participate then a consent form was presented for completion. All participants were given onboarding instructions via telephone call, email and/or in person.

Comments 4. The accessibility and usability testing sections are thorough, but the discussion of the ethical considerations (e.g., patient data security) could be expanded, especially considering the increasing concerns regarding patient-generated health data (PGHD).

Done: line 162; “2.3. Ethical considerations

            Acknowledging the sensitivity of patient-generated health data and participants' desire for robust safeguarding procedures during the stakeholder consultations, several measures were implemented to ensure security and privacy. All data collected through the OverSight app are encrypted both in transit and at rest (converting it into a secure, unreadable format both when being sent and when stored), using industry-standard encryption protocols. Access to participant data was restricted to authorised research team members only, and no personally identifiable information was stored on the app. The onboarding procedure included a detailed explanation of data collection, usage, and storage, and participants were informed of their right to withdraw from the study at any time without any impact on their clinical care”.

Comments 5. Furthermore, the authors could provide more information about the statistical methods used to analyze the data, particularly the correlation analysis between System Usability Scale (SUS) scores and visual acuity.

Done - See line 376.

Comments 6. Were any adjustments made for potential confounders? Including more details on how the app’s user interface was iteratively improved based on the testing feedback would also be helpful.

We engaged users throughout the development process. This included stakeholder consultations before developing the app and iterating the app based on feedback provided by participants across the duration of the pilot. For example, in the manuscript we reported a reduction in the number of steps required to download the app when users experienced more complex onboarding through the TestFlight app which may have affected SUS scores. We also included an illustration (figure 2) showing how the accessibility of surveys was improved by reducing the steps required by the user to complete them and enhancing its use with Apple’s VoiceOver accessibility feature.  The SUS scores will be used to inform future iterations of the app when working towards larger scale studies.

Comments 7. Results: The results are presented clearly, particularly in the breakdown of participant demographics and usability scores. However, while the correlation between visual acuity and SUS scores is statistically significant, the clinical relevance of this finding is not thoroughly discussed.

Done: See lines 394-409; “The SUS scores were within the ‘good’ range, indicating there were no serious issues with usability, but scope for improvements in future iterations of the app. This was demonstrated by the significant negative correlation between BCVA and SUS scores which underscores the impact of visual impairment on app usability. This suggests that users with more severe vision loss may encounter greater challenges when interacting with digital health applications, although other external factors may have come into play. For example, some users who reported average or low usability were those who had to download the TestFlight app before the app was launched on the app store, thereby requiring further steps to onboard. This may also account for the observation that several participants with vision within the better range of the group had borderline or below-borderline usability scores, while one participant with the poorest vision achieved one of the highest usability scores. In addition, the limited sample size may have restricted the statistical power. Therefore, further testing could explore a larger sample size to address other confounding variables, outside of age which had a weak positive correlation, such as type of eye condition or visual impairment.”

Comments 8. It is also worth noting that the sample size of 20 participants is relatively small for drawing strong conclusions about usability and generalizability. More emphasis should be placed on this limitation.

Done: See lines 504-510; “When incorporating digital advancements into OverSight and clinical practice, it is important to also note that varying levels of digital literacy exist, especially amongst the elderly who have the highest risk of eye disease. Although usability testing demonstrated that age didn’t affect usability of the OverSight app, a limitation is that our sample size was small and therefore more work needs to be done on its usability amongst the elderly. If not addressed, it can cause a digital divide and therefore increase inequality rather than solving it.”

Comments 9. Additionally, reporting more detailed findings from the stakeholder consultation could provide greater insight into how specific user needs were addressed in the app's development.

The results of the stakeholder consultation describe 3 main outcomes: survey scheduling, reminders and patient resources. In addition, we address their expectations on safeguarding procedures. Their consideration when integrating into the app is elaborated on in the discussion:

Line 415: “Participants in the workshop gave suggestions for specific app features, such as survey scheduling, reminders and tailored patient resources. These recommendations were actioned and, based on responses to the SUS, appeared to be effective in optimising app design in this early stage of development. These results highlight the value of adopting a co-design process during app development, but also highlight challenges. For example, while providing tailored patient resources for all users is a key target, patients may have complex or rare eye conditions, some with multiple health and ocular comorbidities.

Line 428: “participants also made clear their expectations regarding data security. This was reinforced during the onboarding process, where an explanation was given on how we satisfied safeguarding procedures in the participant information sheet”.

Comments 10. Discussion: The discussion is well-organized and presents a balanced view of the strengths and limitations of the study. The authors successfully argue that the OverSight app could fill a gap in remote ophthalmological monitoring. However, there is room to delve deeper into the potential challenges of integrating such technology into routine clinical practice, such as the digital divide and varying levels of digital literacy among patients.

Done: See lines 504-515;“When incorporating digital advancements like these into OverSight and in clinical practice, it is important to also note that varying levels of digital literacy exist, especially amongst the elderly who have the highest risk of eye disease. Although usability testing demonstrated that age didn’t affect usability of the OverSight app, a limitation is that our sample size was small and therefore more work needs to be done on its usability amongst the elderly. If not addressed, it can cause a digital divide and therefore increase inequality rather than solving it. Even if considered safe and effective, existing challenges remain to ensure interoperability across the healthcare ecosystem. Therefore, for digital health apps like OverSight, ethical considerations and regulatory approvals are ongoing areas of development, necessitating collaboration between technologists, clinicians, and policymakers to establish appropriate guidelines and standards”.

Comments 11. The mention of future developments using Large Language Models (LLMs) is forward-thinking, but speculative; more discussion on the current state of LLMs in healthcare would ground this section better.

Done: See lines 494-504;“Some of these models have multi-modal capabilities in that they can handle various types of biomedical data, including clinical language and medical imaging. The success of LLMs in other medical fields suggests promising potential for their application in ophthalmological care, particularly in processing the rich data generated through a platform like Oversight. For example, it could understand large amounts of PGHD from the app to make predictions on what resources, advice or guidance would be most useful for the patient’s individual circumstances. However, the integration of LLMs into healthcare comes with challenges. Ensuring the accuracy and reliability of AI-generated content is critical, given that errors in medical advice can have serious consequences. There is also the risk of biases in AI models, which can lead to disparities in care”.

Comments 12. Conclusion: The conclusion summarizes the key findings and future directions well. However, it could be strengthened by including more specific next steps for research and development, such as validation studies with larger, more diverse populations, or steps toward regulatory approval for use in clinical practice.

Done: See lines 547-552;“Future research should focus on validating the app with larger and more diverse populations to ensure its usability and effectiveness across different demographics. Additionally, studies assessing the app’s accuracy in detecting clinically significant changes in visual function are essential. Steps toward regulatory approval, including adherence to health data security standards and clinical efficacy trials, will be necessary for integration into routine clinical practice”.

Comments 13. Overall Recommendations:

1. Major Revision: Expand the discussion on ethical considerations regarding PGHD and patient data security. Provide additional details on the stakeholder consultation and how user feedback was integrated into the app’s design.

Done

2. Minor Revision: Improve clarity in the methods section, particularly in describing statistical analyses and selection criteria for participants.

Done

3. Strengths: The paper’s methodology is well-structured, with a clear focus on user-centered design and thorough testing. The authors present a balanced discussion of the app's potential and limitations.

Thank you.

4. Weaknesses: The small sample size limits the generalizability of the results. The novelty of the app compared to existing technologies could be more explicitly emphasized.

Done

Reviewer 3 Report

Comments and Suggestions for Authors

This paper describes the co-design process and development of a smartphone application for digital phenotyping in ophthalmology. The study includes a pilot study and statistical analysis of the results of pilot study.  The project is described in detail and easy to follow.  However, some information is lacking. There are no survey questions provided.

On Page 3, line 103:

 please specify examples of specific features.

On Page 3, line 133

 Please state how many participants/

On Page 4, line 158

It is unclear what is considered  short or long surveys. Please give examples such as the number of questions and types of questions.

 On Page 9, in 305,  the authors stated that

“It was noted that some of the disagreements occurred due tothe initial burden of performing several more steps to download the app through”

What are these disagreements? Please provide more information and what actions were taken

Author Response

Comment 1. On Page 3, line 103:

please specify examples of specific features.

Done: See lines 141-142; “Potential features were discussed such as personalised notifications or reminders and mental health tracking”

Comment 2. On Page 3, line 133

Please state how many participants

Done, see line 175: “3 participants with ultra-low vision or no-perception of light vision”

Comment 3. On Page 4, line 158

It is unclear what is considered  short or long surveys. Please give examples such as the number of questions and types of questions.

Done: See lines 238-243; “The short surveys embedded in the app were the EQ-5D (a standardised measure of health-related quality of life) and the ONS-4 (a concise measure of personal well-being). Longer surveys requiring several more minutes to complete included the Michigan Retinal Degeneration Questionnaire (MRDQ), the National Eye Institute Visual Function Questionnaire (NEI-VFQ), and the Hospital Anxiety and Depression Scale (HADS)”.

 Comment 4. On Page 9, in 305,  the authors stated that

“It was noted that some of the disagreements occurred due to the initial burden of performing several more steps to download the app through”

What are these disagreements? Please provide more information and what actions were taken

Done:

Line 355: “It was noted that some of the disagreements occurred due to the initial burden of performing several more steps to download the app through the TestFlight app however this was later resolved”.

The reference is made to this paragraph:

Line 329: “TestFlight was used for app beta testing which requires users to download an additional app from the Apple store, with further testing instructions before the OverSight app could be downloaded. This was a protracted process and caused onboarding to become unnecessarily complex, particularly when users were not supervised. To overcome this issue, OverSight was submitted to the Apple store for release review and was accepted for public release (Version 1.0 released, 31/08/2023) for assigned participants. As a result, onboarding new participants was a simpler process requiring fewer steps”.

Reviewer 4 Report

Comments and Suggestions for Authors

1. The study is written in a report format rather than an academic study.

2. In the introduction, the contribution of the study to the literature and its importance in terms of literature is not given in a paragraph.

3. In my opinion, it is not possible for the study to contribute to the literature in this form.

4. The study involves the development of a simple mobile application and the interpretation of the results obtained by using it by the users.

5. Currently, such applications have been made or are being made in many areas in the world, even if not in this field.

For all these reasons, I believe that it is not appropriate to accept the study.

Author Response

Comment 1. The study is written in a report format rather than an academic study.

We appreciate your feedback and would like to clarify that the structure of the manuscript is designed to align with the translational nature of our study. This is reinforced by the generally positive comments reported by other reviewers regarding the value of each section of the manuscript. However, in response to your other comments, we have edited the introduction and discussion sections to state more clearly what the existing literature is and how this study fills the gap.

Comment 2. In the introduction, the contribution of the study to the literature and its importance in terms of literature is not given in a paragraph.

Done: See lines 461-478;“Other apps have been developed in the field of ophthalmology which have aimed to improve quality and access to care [59-66]. The general focus of research in this area has been directed at creating app-based versions of tests that would usually take place in clinic, such as visual acuity and contrast sensitivity. Evidence suggests it is feasible to uses these apps for home monitoring, with many demonstrating high reproducibility, accuracy and reliability compared to gold standard assessments [60, 63-66]. Outside of visual acuity, apps such as Alleye have been shown to be effective in detecting metamorphopsia in patients with AMD [64, 65]. However, challenges with adherence is a common issue, highlighting the need for approaches which optimise consistent adoption. Additionally, the responsibility to perform the test accurately is shifted from the provider to the patient. In contrast, OverSight uses passive data already collected in-situ through the smartphone device, thereby reducing the requirement of the user. Unlike apps which predominantly rely on specific task-based assessments, OverSight aims to explore how PGHD can be used as surrogate measures of vision and health status which can be further explored as potential disease biomarkers. In that respect, OverSight offers more than only a single measurement in time and can provide a more holistic approach to understanding the complex dynamics and impact of visual impairment”.

We also updated the introduction to provide a foundation for the points discussed above, focussing on the existing gap in accessible smartphone applications for digital phenotyping in individuals with visual impairment and emphasises how our work addresses this need:

Line 63: “The goal of this approach is to leverage technology to capture details which may be clinically-useful biomarkers of disease (e.g., step counts) [23-26]. However, despite these advancements, there is a notable lack of accessible smartphone applications specifically designed for digital phenotyping in individuals with visual impairment. Existing solutions often do not fully address the usability challenges faced by this population, nor do they effectively utilise PGHD. In this context, we have co-designed and developed the OverSight app, an accessible smartphone application tailored to the needs of people with visual impairment. This approach is particularly pertinent in the context of visual impairment, where eye diseases often progress slowly and insidiously affecting visual function. For instance, as vision worsens patients may display observable changes in behaviour and lifestyle, such as reduced mobility, particularly in low lighting conditions [27, 28]. By focusing on accessibility and user engagement, our study advances the use of digital phenotyping in ophthalmology, offering a novel tool with potential to facilitate remote patient monitoring and inform future care strategies”.

Comment 3. In my opinion, it is not possible for the study to contribute to the literature in this form.

The contribution of our study to the literature is demonstrated in our study through the following points:

  1. We discuss how recent efforts have focused on harnessing digital innovation to revolutionise the assessment and monitoring of visual impairment due to capacity constraints.
  2. We discuss existing solutions such as task-based digital home monitoring (e.g., visual acuity testing) and address how these require active involvement from patients and do not often offer an indication of quality-of-life following diagnosis.
  3. We describe how digital phenotyping could be used to detect clinically significant changes in vision and act as biomarker for disease. Its uniqueness lies in the ability to detect changes passively as opposed to existing measures of remote, smartphone testing.
  4. The objective was to co-design an accessible smartphone app, taking into consideration usability for people with sight impairment.

For this reason, we believe our work does make a valuable contribution to the literature

Comment 4. The study involves the development of a simple mobile application and the interpretation of the results obtained by using it by the users.

This statement is correct – the aim was to design and develop a smartphone application for digital phenotyping in ophthalmology. However, development of health applications like these can be complex and we share how our co-design process resulted in ‘good’ usability according to the SUS scoring thresholds. Plus, we believe that building mobile applications for patients who have visual impairment should be designed in a different way to those targeted with normal vision. The outcome of this study will inform larger scale studies on developing mobile applications in ophthalmology including the feasibility of OverSight as a remote monitoring tool.

Comment 5. Currently, such applications have been made or are being made in many areas in the world, even if not in this field.

This statement is correct. The manuscript has outlined the literature where similar apps have been studied. We analysed digital phenotyping applications that have been used to inform chronic, psychiatric and infectious disease. Although many apps utilise patient generated health data, the study of digital phenotyping in ophthalmology has not been investigated. We reviewed apps that do exist in ophthalmology, outlining their limitations and describe how digital phenotyping could serve as an alternative or complementary measurement. Therefore, this study focusses on how OverSight was developed to advance ophthalmology with digital phenotyping, taking into consideration the needs of visually impaired patients.

Round 2

Reviewer 1 Report

Comments and Suggestions for Authors

Thank you for addressing my suggestions. 

Reviewer 2 Report

Comments and Suggestions for Authors

Authors had answered all of the queries and now it is ready to be accepted.